# Composites Based on Natural Polymers and Microbial Biomass for Biosorption of Brilliant Red HE-3B Reactive Dye from Aqueous Solutions

**DOI:** 10.3390/polym13244314

**Published:** 2021-12-09

**Authors:** Daniela Suteu, Alexandra Cristina Blaga, Ramona Cimpoesu, Adrian Cătălin Puiţel, Ramona-Elena Tataru-Farmus

**Affiliations:** 1Department of Organic, Biochemical and Food Engineering, “Cristofor Simionescu” Faculty of Chemical Engineering and Environmental Protection, “Gheorghe Asachi” Technical University of Iasi-Romania, D. Mangeron Blvd., No. 73A, 700050 Iasi, Romania; 2Department of Materials Science, “Cristofor Simionescu” Faculty of Materials Science and Engineering, “Gheorghe Asachi” Technical University of Iasi-Romania, D. Mangeron Blvd., No. 41, 700259 Iasi, Romania; ramona.cimpoesu@academic.tuiasi.ro; 3Department of Natural and Synthetic Polymers, “Cristofor Simionescu” Faculty of Chemical Engineering and Environmental Protection, “Gheorghe Asachi” Technical University of Iasi-Romania, D. Mangeron Blvd., No. 73A, 700050 Iasi, Romania; puitelac@ch.tuiasi.ro; 4Department of Chemical Engineering, “Cristofor Simionescu” Faculty of Chemical Engineering and Environmental Protection, “Gheorghe Asachi” Technical University of Iasi-Romania, D. Mangeron Blvd., No. 73A, 700050 Iasi, Romania

**Keywords:** biosorption, polymeric composite, reactive dye, *Saccharomyces pastorianus* encapsulated, sodium alginate

## Abstract

Natural polymers have proven to be extremely interesting matrices for the immobilization of microbial biomasses, via various mechanisms, in order to bring them into a form easier to handle—the form of composites. This article aimed to study composites based on a residual microbial biomass immobilized in sodium alginate via an encapsulation technique as materials with adsorbent properties. Thus, this study focused on the residual biomass resulting from beer production (*Saccharomyces pastorianus* yeast, separated after the biosynthesis process by centrifugation and dried at 80 °C)—an important source of valuable compounds, used either as a raw material or for transformation into final products with added value. Thus, the biosorptive potential of this type of composite was tested—presenting in the form of spherical microcapsules 900 and 1500 μm in diameter—in a biosorption process applied to aqueous solutions containing the reactive dye Brilliant Red HE-3B (16.88–174.08 mg/L), studied in a batch system. The preparation and characterization of the obtained polymeric composites (pH_PZC_, SEM, EDS and FTIR spectra) and an analysis of different equilibrium isotherms (Langmuir, Freundlich and Dubinin-Radushkevich—D–R) were investigated in order to estimate the quantitative characteristic parameters of the biosorption process, its thermal effects, and its possible mechanisms of action. The modelling of the experimental data led to the conclusion that the studied biosorption process took place after reaching the Langmuir isotherm (LI), and that the main mechanism was possibly physical, being spontaneous and probably exothermic according to the values obtained for the free energy of biosorption (E = 8.45–13.608 kJ/mol, from the DR equation), as well as the negative values for the Gibbs free energy and the enthalpy of biosorption (ΔH^0^ = −87.795 kJ/mol). The results obtained lead to the conclusion that encapsulation of this residual microbial biomass in sodium alginate leads to an easier-to-handle form of biomass, thus being an efficient biosorbent for static or dynamic operating systems for effluents containing moderate concentrations of reactive organic dyes.

## 1. Introduction

Current technologies for water purification (chemical precipitation, membrane separation, ion exchange, evaporation and electrolysis) have limitations that require the development of new techniques. An important alternative is biosorption—a cost-effective, simple, reversible, passive accumulation process by which inactive biosorbent binds (through ion exchange, absorption, adsorption and surface complexation) certain ions or molecules from aqueous solutions. The advantages of biosorption include: simplicity, no nutrient requirements for the non-living biomass, low sludge generation, low operational costs and high efficiency [1,2,3,4,5,6].

Microorganisms are, in biotechnology, important sources for a great variety of intracellular and extracellular compounds such as: organic acids, amino acids, antibiotics [1]. In these biosynthetic processes, the residual microbial biomass is an inevitable waste generated in the separation step. These by-product microorganisms (bacteria, yeast or fungi), could be used as a potential alternative to existing technologies for the recovery of pollutants from industrial waste streams, due their ability to retain, by different mechanisms, pollutants from aqueous streams through biosorption.

Microbial cells have a high surface area-to-volume ratio and can thus provide a large contact surface, but, most importantly, they contain in their cell walls and plasma membranes several natural polymers with many functional groups (such as carboxyl, hydroxyl, amine, imidazole, phosphate, sulfhydryl and sulphate groups), which can bind the contaminant during the sorption process [7].

The use of microorganisms in environmental depollution technologies can be carried out with them either in a free form or immobilized on polymeric supports [1]. For the biosorbents obtained by microorganism immobilization (which offers several advantages such as easy separation, enhanced operational stability, multiple uses, being incorporable into fixed and fluidized bed columns and higher productivity), different natural or synthetic polymers can be used for the production of inexpensive, non-toxic carriers with reactive functional groups [3,4]. From these, natural polymers include chitosan, alginate, agar, collagen and agarose, and synthetic polymers include poly-acrylamide, polyvinyl alcohol, polyethylene-glycol, polypropylene, polyethylene, polyvinylchloride, poly-urethane and polyacrylonitrile [3,4]. Natural polymers are preferred in the separation processes due to their biocompatibility and the possibility of their production from renewable sources, but some limitations regarding their stable structure, which is source-dependent, are known. Synthetic polymers have more stable and controllable structures and functionalities, completed by high chemical and biological stabilities [8,9]. Different immobilisation techniques can be applied to a microbial biomass: entrapment, cross-linking, covalent bonding, adsorption and encapsulation. The main problem related to the use of microencapsulation for biosorbent production is the limitations imposed by the diffusion step in the pollutant retention [4].

Different microorganisms have been used successfully in environmental pollutant removal:Bacteria: *Pseudomonas aeruginosa* immobilised in sodium alginate have been used for the retention of Reactive Green 6 from wastewaters with a maximum adsorption capacity of 21.2 mg/g; *Bacillus* sp., immobilized in 1% sodium alginate, allows the obtaining of a maximum adsorption capacity of 588.235 mg/g for Brilliant Red HE-3B; *Bacillus cereus* immobilised in 3% sodium alginate yields a maximum retention of 83% for Malachite Green [10,11,12].Fungi: *Penicillium* sp. immobilised in 2% sodium alginate has been used for the removal of C.I. Reactive Red with a maximum adsorption capacity of 120.48 mg/g; *Penicillium crustosum* immobilised in 2% agar retains Congo red with an efficiency of 81.86%; *Rhizophus orizae* immobilised in carboxymethyl cellulose has been used for the biosorption of Reactive Blue with an efficiency of 97.44%. *Saccharomyces cerevisiae* immobilized in sodium alginate and present in the form of gel beads has also been used for the retention of Brilliant Red HE-3B dye, and leads to a sorption capacity of 104.67 mg/g [13,14,15,16].

Biosorbents obtained through immobilization using non-toxic biopolymers may help improve biomass biosorption capacity and facilitate its separation from wastewater solutions [8]. Pires et al. obtained an increase in the biosorption abilities of *Cupriavidus*, *Sphingobacterium* and *Alcaligenes* through immobilization in naturally occurring (alginate and pectate) and synthetic polymers (synthetic cross-linked polymer) of up to 12-fold when compared to the use of the polymers alone [7]. Khashei et al. investigated the biosorption ability of heavy metals of immobilized *Pseudomonas putida* cells in various matrices (alginate–PVA–CaCO_3_ and carboxymethyl cellulose). An increase in metal removal efficiency in all matrices after bacterial immobilization was observed: 75.5% Pb(II) compared to 60% and 75% Cd(II) to 20% without cells for alginate–PVA–CaCO_3_, and 32% Pb(II) retention to 3% and 15% Cd(II) to 5% with cellulose support [17].

*Saccharomyces pastorianus* (*Saccharomycetaceae* family) is an unicellular yeast, characterized by its ability to convert, enzymatically, sugar into carbon dioxide and alcohol, and is a by-product in the brewing industry. This residual biomass could be used as a potential alternative to existing technologies for the recovery of pollutants from industrial waste streams. The cell walls of *Saccharomyces pastorianus* comprise of mannoproteins and β-glucans, highly entangled in the cell wall matrix; *N*-linked type mannoproteins, being the majority of yeast mannoproteins, are composed of 90% carbohydrate and 10% protein, providing phosphate groups such as mannosylphosphate residues that contribute to the ionic properties of the yeast cell surface and can act as biosorption functional groups [18]. Bastos et al. observed an increase in yeast flocculation and an increase in the negative charge of the yeast surface after the brewing process, which could suggest that residual *S. pastorianus* could be a better biosorbent than the initial strain [19].

The aim of this paper is to investigate the biosorptive properties of a newly proposed polymeric composite based on a residual microbial biomass of *Saccharomyces pastorianus* encapsulated in sodium alginate. For this, we approached a working protocol that aims at three stages: (1) preparation and physical–chemical characterization of a prepared polymeric composite based on a residual biomass of *Saccharomyces pastorianus* encapsulated in sodium alginate; (2) an investigation of the influence of certain physical parameters on the study of the biosorption process, such as the dose of the biosorbent, the size of the composite granules, the pH of the solution, the initial concentration of the dye solution, and the temperature; (3) processing of experimental data using different adsorption equilibrium isotherms in order to estimate the characteristic parameters and thermal effects of the studied bioprocess for the treatment of dye-containing watery effluents.

## 2. Materials and Methods

### 2.1. Materials

*Biomass. Saccharomyces pastorianus* are members of the family *Saccharomycetaceae*. The residual biomass *S. pastorianus* (an interspecies hybrid of *Saccharomyces cerevisiae* and *Saccharomyces eubayanus* [18]) was provided by a local brewing company (Albrau, Onesti, Romania). The residual biomass was separated by centrifugation (8000 rpm), dried at 80 °C and then microencapsulated in sodium alginate (Figure 1a).

*Biosorbent.* The prepared polymeric composites based on residual biomass granules (*Saccharomyces pastorianus*) used in the biosorption experiments were obtained by microencapsulation using a BUCHI B390 microencapsulator. The suspension necessary for obtaining the granules was prepared from 1.5% low-viscosity grade sodium alginate (Figure 1a) purchased from Buchi Labortechnik AG (Flawil, Switzerland) (prepared in distilled water at 70 °C) and 5% residual biomass. The nozzles diameters used were 450 and 750 μm, with the following conditions: air pressure 100 mbar, T = 45 °C, 500 V and 800 Hz (for 750 μm) and 200 Hz (for 450 μm), allowing the production of beads with 900 and 1500 μm diameters. The suspension was dripped into a 100 mM calcium chloride solution (prepared in distilled water at 5 °C), to obtain spherical beads with Φ1 = 900 μm/Φ2 = 1500 μm diameters. The schematic representation of all the steps involved in the preparation of the biosorbent with *Saccharomyces pastorianus* is shown in Figure 1b. All granules of the polymeric consortium residual biomass–sodium alginate had a uniform size (Figure 1c) and were stable, without adhesion phenomena manifesting between them during storage in an aqueous solution of calcium chloride 10 mM at a temperature of 5 °C.

*Adsorbate*. A reactive dye, Brilliant Red HE-3B (BRed-C.I. 25810; MW = 1430 g/mol, λ_max_ = 530 nm, Bezema Colour Solutions, Montlingen, Switzerland) with the chemical structure shown in Figure 2a, was selected as the chemical pollutant (the reference model of a reactive dye) of the aqueous system for this study. A stock solution (with a concentration of 500 mg dye/L) was prepared using a commercial salty form of the dye of analytical reagent purity grade, and distilled water. For experiments, solutions were prepared from the stock solution by appropriate dilution with distilled water.

Other chemicals used in experiments were of analytical purity, were used without further purification and were purchased from Chemical Company, Iasi, Romania

### 2.2. Methods

#### 2.2.1. Batch Biosorption Methodology

The batch biosorption studies were performed using 50 mL Erlenmeyer flasks, in which there were introduced different amount of encapsulated biomass with 5% dry matter (dw), pre-washed with distilled water to remove traces of calcium chloride solution that would cause dye precipitation, as well as 25 mL of dye solution at different initial concentrations (in the range of 16.88–174.08 mg/L). The pH values were adjusted with a 1 N HCl solution, and the constant desired temperatures (5°, 30°, 45 °C) were ensured using a thermostatic bath with a contact time of solid–aqueous phases of about 24 h (Figure 2b). After reaching equilibrium, the dye content in the supernatant was spectrophotometrically determined using a Shimadzu UV-1280 UV-VIS Spectrophotometer (Shimadzu Corporation, Kyoto, Japan) at the maximum dye wavelength of 530 nm.

The biosorption capacity of the prepared biosorbent (q, mg of dye/g of biosorbent) was calculated using Equation (1):(1)$$q=\frac{C_{0}-C}{G}\cdot V$$
where C_0_ and C are the dye’s initial and equilibrium (residual) concentrations in solution (mg/L), G is the amount of biosorbent (dry matter (d.w.) from alginate granules; g) and V is the volume of solution (L).

#### 2.2.2. Physicochemical Characterization of Composite Biosorbent

The characterization of the prepared polymeric composite, in order to highlight the internal structure and the functional groups responsible for the biosorbtive properties, was made using physico-chemical methods (SEM, FTIR) before and after the biosorption process.

Lyophilization. Prior to the SEM investigation, the composite granules were lyophilized using an equipment Labconco lyophilizer (Labconco, Kansas City, MO, USA). SEM images were recorded with a HITACHI SU 1510 (Hitachi SU-1510, Hitachi Company, Tokyo, Japan) Scanning Electron Microscope, MNPs were fixed on Aluminum stubs and coated with a 7 nm-thick gold layer using a Cressington 108 (Cressington Scientific Instruments Ltd., Watford, UK) device before observation.

Scanning Electron Microscopy (SEM) was carried out to evaluate the surface micromorphology of the composite polymeric materials based on *Saccharomyces pastorianus* encapsulated in sodium alginate before and after the biosorption process. A scanning electron microscope VegaTescan LMH II (Tescan Orsay Holding, Brno – Kohoutovice, Czech Republic), detector SE: WD 15.5 mm, 30 kV, HV; VegaTC software with an EDS detector XFlash 6/10 Bruker (Bruker, Karlsruhe, Germany), automatic mode and mapping distribution of elements; and Esprit 2.2 software (Bruker, Karlsruhe, Germany) were used.

Fourier transform infrared (FT-IR) was applied in order to identify the functional groups existing in the initial biosorbent, as well as those involved in the biosorption process, respectively, in the binding of dye molecules. For this, FTIR spectra were registered using a Bruker Vertex 70 FT-IT spectrophotometer (Bruker, Karlsruhe, Germany) in total attenuated reflectance mode in the wavenumber range 4000–400 cm^−1^, with a resolution of 2 cm^−1^ and 32 acquisitions at room temperature.

#### 2.2.3. Modelling the Biosorption Experimental Data

For experimental data modelling, three of the most-known biosorption equilibrium models: Freundlich (F), Langmuir (L) and Dubinin–Radushkevich (D–R), briefly explained below, were applied [20].

The **Freundlich model (F)** takes into consideration the surface heterogeneity and exponential distribution of the active sites of the biosorbent. The nonlinear/linear forms of the equation are:(2)$$q=K_{F}\cdot C_{}^{1/n}/\log q=\log K_{F}+\frac{1}{n}\log C_{}$$
where K_F_ and 1/n are constants associated with the biosorption capacity and intensity (efficiency), respectively; a favorable biosorption corresponds to a value of 1 < n < 10.

The **Langmuir model (L)** considers a monolayer distribution of the solute molecules on the biosorbent surface, which contains a finite number of energetically equivalent sites. The linearized form of Equation (3) can be described by the following two forms: L1 (4) and L2 (5):(3)$$q=\frac{K_{L}\cdot C\cdot q_{0}}{1+K_{L}^{}\cdot C}$$
(4)$$L1:\;\frac{1}{q}=\frac{1}{q_{0}}+\frac{1}{K_{L}\cdot q_{0}}\cdot \frac{1}{C}$$
(5)$$L2:\;\frac{C}{q}=\frac{1}{q_{0}\cdot K_{L}}+\frac{C}{q_{0}}$$
where q_0_ is the maximum amount of sorbed solute (mg/g) and K_L_ is the constant related to the binding energy of solute (L/mg).

The **Dubinin–Radushkevich model (D–R)**, through the parameter that is determined—the E-energy of the biosorption process—allows the characterization of a process’s nature (physical or chemical). A value for E higher than 8 KJ/mol suggests a physical biosorption mechanism and values between 8 and 16 KJ/mol indicate an ion-exchange mechanism. The nonlinear/linear forms of the characteristic equation are:(6)$$q=q_{0}\exp\left(-B\cdot \varepsilon^{2}\right)/\ln\;q=\ln\;q_{0}-B\varepsilon^{2}$$
$$\varepsilon =\mathrm{RT}\;\ln\left(1+\frac{1}{C}\right)/E=\frac{1}{\sqrt{2B}}$$
where q_D_ is the maximum biosorption capacity (mg/g); B is the activity coefficient related to the mean biosorption energy; ε is the Polanyi potential and E is the mean free energy of biosorption (kJ/mol)

### 2.3. Thermodynamic Parameters of the Biosorption Process

Based on the value of the Langmuir constant, K_L_, and temperature, three thermodynamic parameters were determined using the following equations [21,22]:(7)$$\Delta G=-RT\ln K_{L}$$
(8)$$\ln K_{L}=-\frac{{\Delta H}^{0}}{\mathrm{RT}}+\frac{{\Delta S}^{0}}{R}$$
where ΔG is the free energy (kJ/mol), ΔH is the enthalpy (kJ/mol) and ΔS is the biosorption entropy change (kJ/mol K); R is the universal gas constant (8.314 J/mol K), T is the absolute temperature of the solution (K) and K_L_ is the value of the Langmuir constant (L/mol).

## 3. Results and Discussion

### 3.1. Analysis of the Biosorbent Based on Residual Biomass Using SEM, EDAX and FT-IR Spectra

Scanning electron microscopy was used to study the morphology of composite granules obtained from the residual biomass of Saccharomyces pastorianus encapsulated in sodium alginate and, also, information about pore distribution and locations. Images obtained at 25, 500 and 1000× magnifications are shown in Figure 3.

Energy-dispersive X-ray spectroscopy (EDS, EDAX, EDX, EDXS or XEDS) techniques were used for the elemental analysis and chemical characterization of the samples, representing the polymeric composites tested in this study, as biosorbents—before and after the biosorption process of Brilliant Red dye HE-3B.

The SEM images presented in Figure 3a easily show the mesoporous appearance of the analysed material. Figure 3b shows the SEM images of the composites after the biosorption process of the Brilliant Red HE-3B dye, showing a slight change in the appearance of the granules’ surface, which became more uniform. Porosity and a large surface area are very important elements in the case of materials used as adsorbents for chemical pollutants.

The EDAX spectrum obtained for the analyzed samples showed, on the surfaces, the presence of various elements that come from the structure of polymeric composites based on microbial biomass and sodium alginate (Figure 3a) and also of the retained dye, respectively the increase of carbon and appearance of nitrogen and sulfur atoms (Figure 3b). These modifications suggest the retention of the dye on the surface of the polymeric composite material. The difference observed in the amounts of calcium before and after biosorption could be due to the process of granule washing with distilled water carried out before the biosorption process.

The FTIR spectra obtained for the biosorbent before and after the dye biosorption process, as well as the spectrum of the studied dye Brilliant Red HE-3B, are presented in Figure 4.

A study of the FTIR spectra from Figure 4 highlights the following aspects:

(1) Peaks that are present in both types of samples—before and after the biosorption process—are characteristic of the polymer–biomass composite, and come from either the structure of the sodium alginate or from that of the biomass. Characteristic in this regard are the peaks recorded at 3400–3290 cm^−1^, which are specific for the O–H group, while the sharp peaks at 2950 and 2850 cm^−1^ can be assigned to C–H stretching; 1050–1125 cm^−1^ can be assigned to C–O stretching vibrations characteristic of the alcohol structures; 1014 cm^−1^ can be assigned to CO stretches in the polysaccharides; 1600 cm^−1^ can be assigned to aromatic ring stretching; and 1400 cm^−1^ can be assigned to OH bending, which has reduced intensity in samples of biosorbent after biosorption.

(2) Peaks that appear in the biosorbent sample after dye retention in the area of 600–1800 cm^−1^ provide a valuable clue that suggests the dye’s retention. The most obvious in this sense is the peak at 1325 cm^−1^, characteristic of sulfonic acid sodium salts, which are also found in the dye spectrum. Other peaks found in the dye spectrum and in the biosorbent after biosorption are: 1727 and 1540 cm^−1^ assigned to C=O; 1625 cm^−1^ assigned to aromatic ring stretching; 1485 cm^−1^ assigned to C=O; and 1210 cm^−1^ assigned to COC.

### 3.2. Evaluation of the Value of the Point of Zero Charge (pH_PZC_) for Biosorbent

A parameter that allows the appreciation of the electric charge of the biosorbent surface, and therefore its behaviour during interactions with different chemical species—anionic and/cationic—is the point of zero charge (pH_PZC_). The value of pH_PZC_ (pH of zero charge) for biosorbent based on residual biomass immobilized in sodium alginate was determined using the method proposed by Nouri and Haghseresht, following the graphical representation in Figure 5 [23].

The pH_PZC_ value was found to be 5.4 (Figure 5). At values of pH < pH_PZC_, the composite surface was positively charged due to the increased H+ ion concentration (the characteristic groups were positively charged) and susceptible to react with anionic species via electrostatic interactions and hydrogen bonding. At pH > pH_PZC_ values, the composite surface was negatively charged due to the dissociation of some characteristic functional groups and was capable of ion-exchanges and/or electrostatic interactions with cationic species.

### 3.3. Modelling the Biosorption Equilibrium Process

Our previous study had concluded that the biosorption process of Brilliant Red HE-3B dye onto residual biomass of *Saccharomyces pastorianus* encapsulated in sodium alginate proceeded with adequate results under the following conditions: pH = 3, contact time of 24 h, temperature of 25 °C and the concentration of biosorbent in the range of 0.06 to 0.08 g (with 5% d.w.), depending on the diameter of the biomass-based granules [24].

Thus, this study focuses on the analysis of the biosorption balance of the Brilliant Red HE-3B reactive dye on residual biomass-based biosorbent prepared by the encapsulation of *Saccharomyces pastorianus* (residual yeast biomass) in sodium alginate. This involves determining the quantitative characteristic parameters that describe the process, determining thermodynamic parameters, and evaluating the biosorption mechanism. The proposed equilibrium isotherms, presented in Figure 6, were processed using three of the most well-known isotherm models in the scientific literature (Equations (2)–(8)) in order to determine their characteristic quantitative parameters.

The isotherms presented in Figure 6 show that retention on smaller granules (Φ2, Figure 6b) was better compared to that obtained when larger granules were used (Figure 6a), at all temperatures at which the biosorption process was studied for the considered reactive dye. The alignment of the curves indicates a type of “L” isotherm, subgroup 2, according to Giles classifications [25]. This is the classic Langmuir-type isotherm which is based on the surface biosorption of vertically oriented molecules through particularly strong intermolecular bonds. The experimental data were modeled by applying the linearized forms of each proposed biosorption model (Freundlich, Langmuir I and II and Dubinin–Radushkevich, Equations (2)–(8)). The obtained graphs are presented in Figure 7 and the results, calculated according to the intercepts and slopes of the corresponding linear plots, are listed in Table 1.

Assessment of the best isotherm model to describe the biosorption process of the reactive Brilliant Red HE-3B dye on the biosorbent based on encapsulated residual biomass of *Saccharomyces pastorianus* in sodium alginate took into account mainly the values of the correlation coefficients: R^2^.

The processing of the equations of the lines in Figure 7 led to the parameters characteristic of the biosorption process, systematized in Table 1.

The analysis of the data obtained by processing the experimental results (Table 1) based on the three models considered, allowed the evaluation of some quantitative parameters necessary for:-Assessment of the efficiency of polymeric composites based on residual biomass as bioadsorbent material (q),-Subsequent analyses to evaluate the effect of temperature on the development of the biosorption process and its feasibility from a thermodynamic point of view (q, K_L_).-Additionally, a series of preliminary conclusions can be drawn regarding the mechanism of the biosorption process, both from the information provided by the value of the biosorption energy (E) determined from the DR model, and from the subsequent data obtained by the thermodynamic study based on the Langmuir model: biosorption capacity, q and Langmuir constant, K_L_.

Comparatively analysing the values of the correlation factors, R^2^, from Table 1, it is observed that the model that best confirms the experimental data is Langmuir, with the L1 data representation form.

Studying the values of the calculated parameters, an important influence of temperature was observed. The best results for biosorption capacity, q, and the Langmuir constant, K_L_, were obtained at a temperature of 30 °C. Above this temperature, the values obtained were slightly lower—one explanation being the instability of the microbial biomass, which leads to its denaturation at higher temperatures.

The free biosorption energy (E), calculated by the D–R model, provides some preliminary information on the nature of the biosorption process (physical or chemical). The values obtained were around 8–9 KJ/mol—values that characterize adsorption processes based on physical bonds (van der Waals, hydrogen, dipole–dipole interactions and electrostatic attraction between the positively charged surface of the biosorbent and the anionic functional groups in the dye molecules, respectively) between the dye and the polymeric composite based on the *Saccharomyces pastorianus* residual biomass encapsulated on sodium alginate as biosorbent.

The value obtained for the biosorption capacity under ambient temperature conditions (30 °C), according to the Langmuir I model, was comparable with other values of biosorption capacity published in the literature for different types of microbial biomass in free or immobilized/encapsulated forms on various polymeric matrices in order to remove organic dyes from aqueous media. For the removal of Reactive Red 198 and Reactive Yellow 2 dyes, a microbial cell-immobilized *Platanus orientalis* leaf tissue was used and biosorption capacities of 51.12 and 37.59 mg/g were recorded [26]. Immobilized *Mucor plumbeus* on a sepiolite support was used for Methyl Violet biosorption, with 187.76 mg/g biosorption capacity [27]. *Penicillium* sp. immobilized in 2% sodium alginate was used for the removal of C. I. Reactive Red dye, obtaining a maximum sorption capacity of 120.48 mg/g [13]. *Lentinus concinnus* biomass, immobilized to carboxymethyl cellulose (CMC) in the presence of FeCl_3_, achieved the removal of Disperse Red 60 dye with a maximum q value of 92.6 mg/g [28]. The same strain (*Lentinus concinnus*) immobilized in polyvinyl alcohol/polyethylene oxide hydrogels was employed for the retention of Reactive Yellow 86 dye, with a biosorption capacity of 87.6 mg/g [29]. *Bacillus subtillis* sp. immobilized in sodium alginate was used for Brilliant Red HE-3B reactive dye biosorption, obtaining a 588.235 mg/g maximum sorption capacity [30]. *Lactobacillus* sp. immobilized in sodium alginate successfully removed Orange 16 dye from aqueous solutions, with a q = 123. 459 mg/g value [31].

### 3.4. Analysis of the Proposed Thermodynamic Parameters

Three thermodynamic parameters were calculated (Table 2), using Equations (2) and (3) [32,33]. In these equations, the values of the Langmuir constant (Ll) K_L_ (L/g) were taken into account in the case of Φ1 = 1500 μm granules, for which the best regression coefficients and higher values for the biosorption capacity were achieved.

According to the data from Table 2 of Gibbs free energy, ΔG^0^ shows negative values (except at a temperature of 5 °C), which suggests that overall, the biosorption of the reactive dye Red Brilliant HE-3B on the polymeric composite tested as biosorbent (based on *Saccharomyces pastorianus* residual bacterial biomass encapsulated in sodium alginate) could be considered a spontaneous process. Additionally, ΔG^0^ values between −20 and 0 KJ/mol indicate a biosorption process based on a physical adsorption mechanism. This finding overlaps with the preliminary data provided by the values of the average free adsorption energy (E), calculated using the equation of the DR model.

The negative value resulting from the calculations for the biosorption enthalpy (ΔH^0^) (slope of the linear dependence of ln K_L_ vs. 1/T), confirms both the exothermic nature of the biosorption of the reactive dye Brilliant Red HE-3B on the studied biosorbent, and the more physical nature of the mechanism of the biosorption process [29].

The positive value of the adsorption entropy (ΔS^0^) characterizes the increased randomness at the solid–solution interface during the dye biosorption process and some structural changes of the biosorbent and adsorbant.

### 3.5. Recovery of Biosorbent Loaded with Dye

Due to its “fragile” nature, the biosorbent cannot be regenerated by recovering the dye and reusing it in new adsorption–desorption cycles. The recovery can be performed in the direction of use as a bioamendment for the soil, in order to improve the quality of the soil [34] in composting or anaerobic digestion processes [35].

## 4. Conclusions

The results here confirm that residual microbial biomasses encapsulated in a polymeric matrix—sodium alginate in this study—can be considered biomaterials with biosorbent properties, effective in retaining organic dyes present in aqueous solutions in moderate concentrations. The biosorption process of the reactive dye Brilliant Red HE-3b on polymeric composites of different granulations, obtained by encapsulating the residual biomass of *Saccharomyces pastorianus* in the sodium alginate matrix, was studied. For this purpose, the modelling of the experimental data was performed using a series of known isothermal models: Langmuir, Freundlich and Dubinin–Radushkevich, and it was proved that the Langmuir model is the one that fits best in this regard.

The study showed that this biosorption process, in which a biosorbent based on *Saccharomyces pastorianus* encapsulated on sodium alginate for the removal of the reactive dye Brilliant Red HE-3B is used, is of a physical nature, according to the calculated value of the free adsorption energy (E = 8.45–13.608 kJ/mol, from the DR model equation) and the value of the biosorption enthalpy (ΔH^0^ = −87.795 kJ/mol). Additionally, the negative values of its free energy (ΔG^0^ = −5903 and −12,111 kJ/mol) and the negative enthalpy of biosorption (ΔH^0^ = −87,795 kJ/mol) suggested that the process could be considered spontaneous, and probably of an exothermic nature.

## Figures and Tables

**Figure 1 polymers-13-04314-f001:**
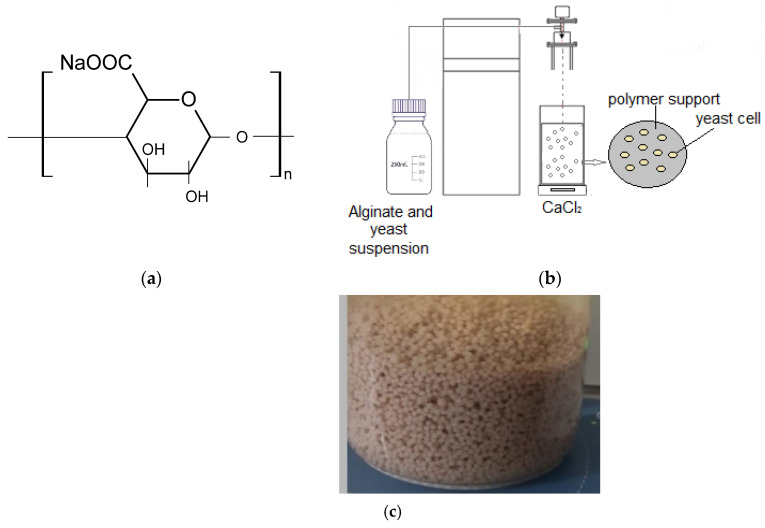
(**a**) Chemical structure of the polymeric matrices for encapsulation; (**b**) Schematic representation of the microencapsulation process; (**c**) Encapsulated biomass granules.

**Figure 2 polymers-13-04314-f002:**
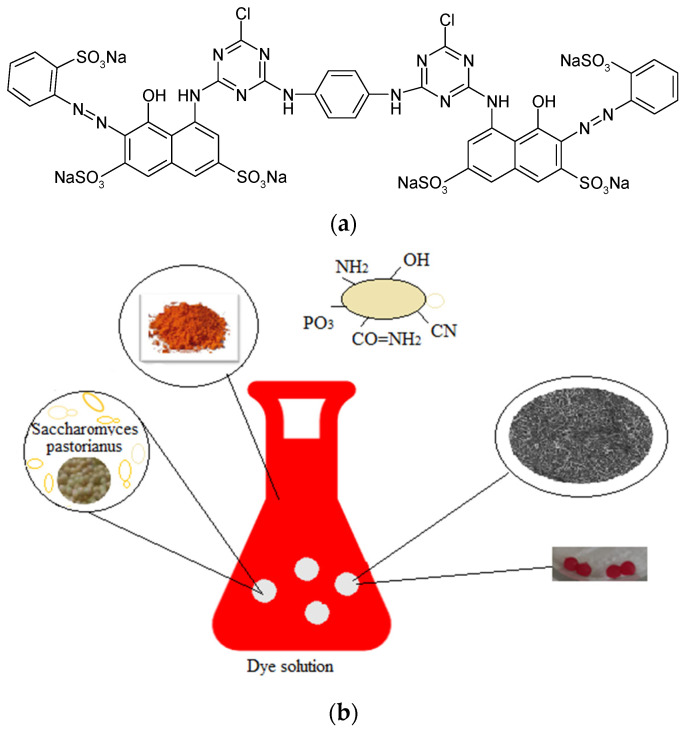
Schematic representation of the biosorption process of Brilliant Red HE-3B (**a**) on a polymeric composite based on *Saccharomyces pastorianus* residual biomass encapsulated in sodium alginate (**b**).

**Figure 3 polymers-13-04314-f003:**
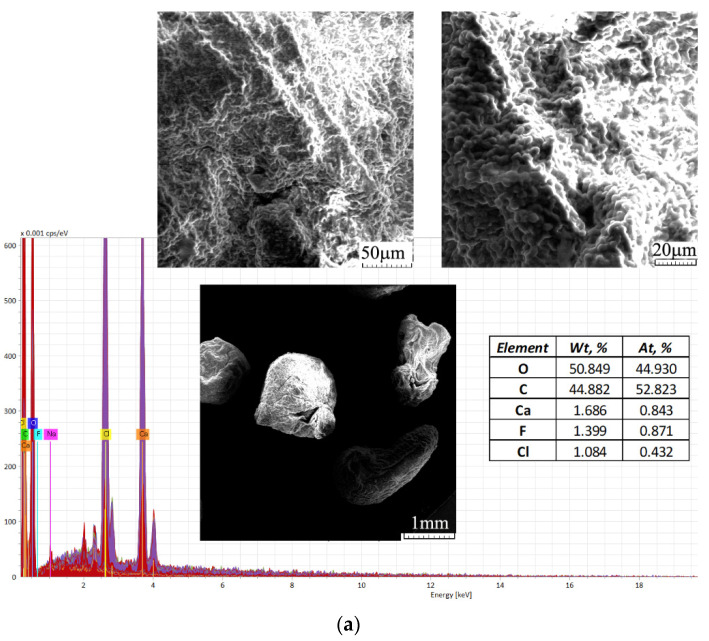
Scanning electron microscopy (SEM) and energy-dispersive X-ray (EDX) spectrum of the polymeric composite before (**a**) and after (**b**) Brilliant Red HE-3B dye biosorption.

**Figure 4 polymers-13-04314-f004:**
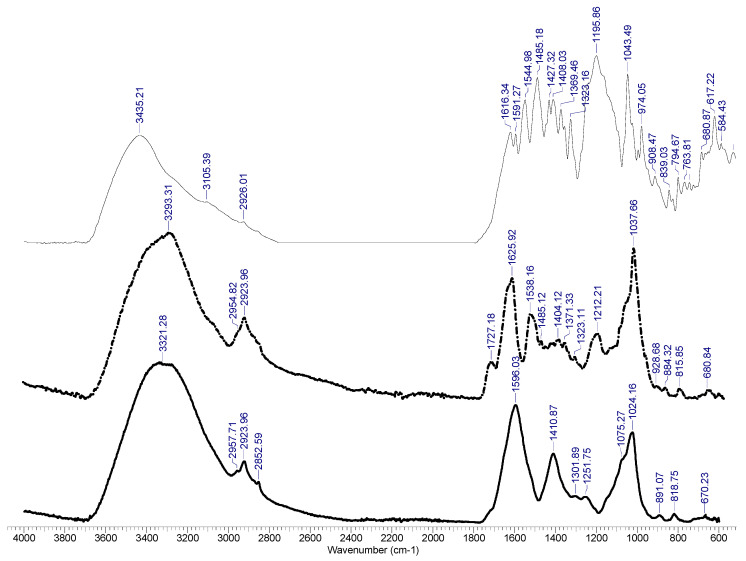
FT-IR spectra of the polymeric composite before (**-** the continuous bold black line) and after (--- discontinuous black line) biosorption, and for the Brilliant Red HE-3B dye (- the continuous black line).

**Figure 5 polymers-13-04314-f005:**
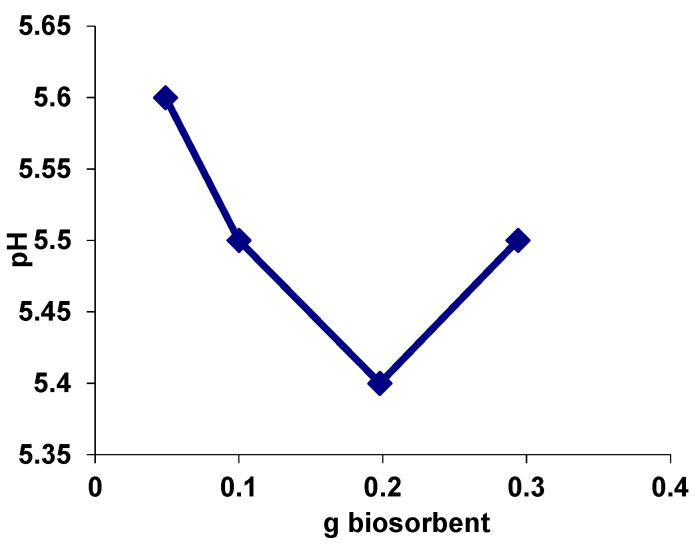
The value of the point of zero charge (pH_PZC_) for biosorbent based on residual biomass of *Saccharomyces pastorianus* encapsulated in sodium alginate.

**Figure 6 polymers-13-04314-f006:**
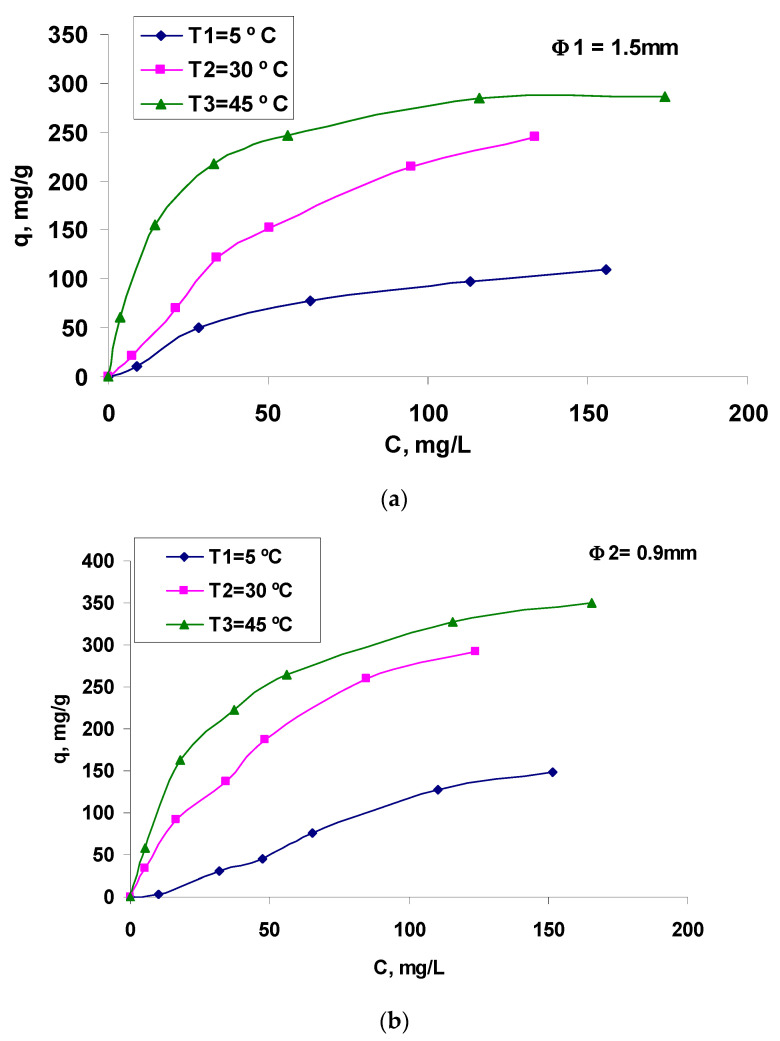
Biosorption isotherms of Brilliant Red HE-3B reactive dye on composite material based on residual *Saccharomyces pastorianus* biomass encapsulated in sodium alginate, in the form of two-dimensional granule: Φ1 = 1500 μm (**a**) and Φ2 = 900 μm (**b**). Conditions: pH = 3, contact time = 24 h, dry substance content in biosorbent = 0.16/0.14 g/L.

**Figure 7 polymers-13-04314-f007:**
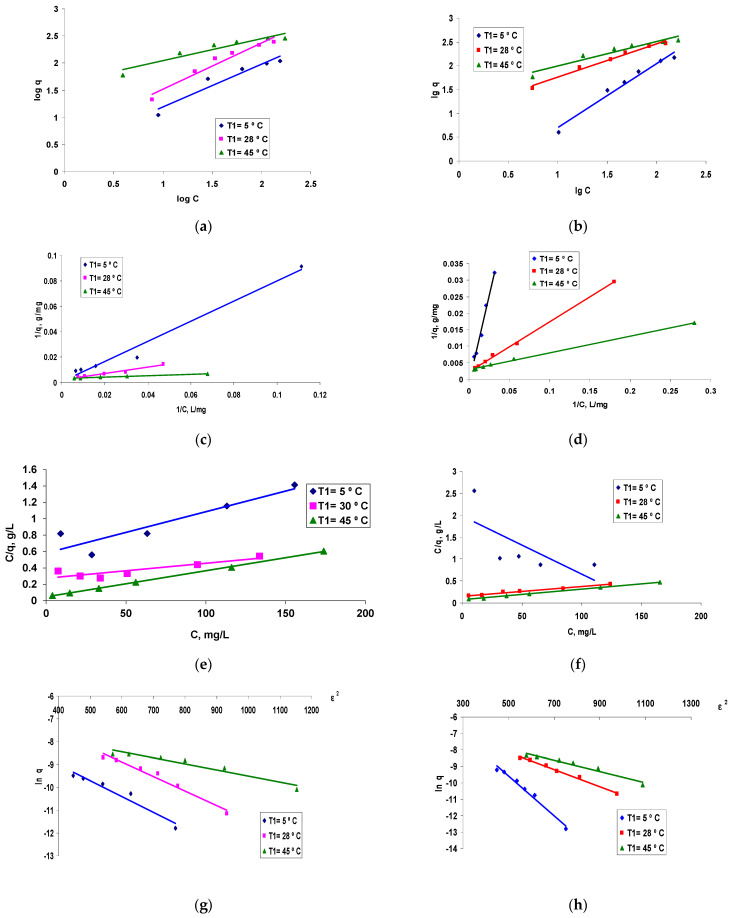
Linearized forms of Freundlich (**a**,**b**), Langmuir I (**c**,**d**), Langmuir II (**e**,**f**) and DR (**g**,**h**) plots for the Brilliant Red HE-3B reactive dye on composite material based on residual *Saccharomyces pastorianus* residual biomass encapsulated in sodium alginate. Conditions: pH = 3, contact time = 24 h, dry substance content in biosorbent = 0.16/0.14 g/L; diameter of granules: Φ1 = 1500 μm (**a**,**c**,**e**,**g**) and Φ2 = 900 μm (**b**,**d**,**f**,**h**) at three temperatures: 5°, 30° and 45 °C.

**Table 1 polymers-13-04314-t001:** Characteristic parameters for the biosorption of Brilliant Red HE-3B reactive dye onto composite material based on residual biomass of *Saccharomyces pastorianus* encapsulated in sodium alginate.

Isotherm	Φ1 = 1500 μm			Φ2 = 900 μm		
	278 K	303	318	278 K	303	318
Freundlich						
K_F_ ((mg/g) (L/mg)^1/n^)	2.5275	4.6709	43.944	0.222	11.416	30.4369
n	1.2711	1.1737	2.4845	0.7402	14.431	1.9558
R^2^	0.9242	0.9525	0.9025	0.971	0.9838	0.9321
Langmuir I (1/q = f (1/C))						
q_0_ (mg/g)	2500	555.55	312.5		454.545	344.827
K_L_ (L/g)	0.000504	0.00712	0.0667		0.01453	0.0571
R^2^	0.9799	0.9761	0.9987	0.9881	0.9985	0.9969
Langmuir II (C/q = f (C))						
q_0_ (mg/g)	200	555.55	312.5		476.19	416.667
K_L_ (L/g)	0.00859	0.00656	0.06639		0.01345	0.03162
R^2^	0.8363	0.7778	0.9994	0.4894	0.9838	0.9991
Dubinin-Radushkevich (DR)						
q_0_ (mg/g)	3013.02	8814.576	1572.725	42,327.8	5348.6	2923.39
β (mol^2^/KJ^2^)	0.007	0.0063	0.0027	0.0121	0.0051	0.0034
E (KJ/mol)	8.4515	8.909	13.608	6.428	9.901	12.127
R^2^	0.9426	0.9724	0.9317	0.9827	0.9917	0.954

**Table 2 polymers-13-04314-t002:** Thermodynamic parameters for the biosorption process of reactive Brilliant Red HE-3B dye onto composite based on residual biomass of *Saccharomyces pastorianus* encapsulated in sodium alginate.

T (K)	K_L_ (L/g)	ΔG^0^ (KJ/mol)	ΔH^0^ (KJ/mol)	ΔS^0^ (J/mol K)
278	0.0000504	−0.704	−87.795	312.232
303	0.00712	−5.903		
318	0.0667	−12.111		

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
