# Peer review of "Composites Based on Natural Polymers and Microbial Biomass for Biosorption of Brilliant Red HE-3B Reactive Dye from Aqueous Solutions"

_polymers, 2021, doi:10.3390/polym13244314_

Round 1
Reviewer 1 Report
In their manuscript, the authors characterise biocomposites from microbial brewery residues immobilised in alginate and determined the adsorption isotherms of the reactive dye Brilliant Red HE-3B using three commonly used models (Freundlich, Langmuir and Dubinin-Radushkevich). This approach of using biological waste products from industrial processes is promising from a sustainability point of view and is already well documented in the relevant literature by many other examples. The manuscript is mostly well structured and the description largely consistent. However, the manuscript has some shortcomings that stand in the way of publication in its current form. How, for example, did the authors verify the complete inactivation of the yeasts by drying? Was the biomass used characterised in more detail, for example the tendency to form spores?
Furthermore, several orthographic and grammatical errors or incorrectnesses are conspicuous in the text, even for the linguistically unskilled reader. The presentation of results needs to be revised, as does the list of literature references. The following comments are intended to help with this:
Line 33: please correct … microcapsules of about 1500 µm
Line 75: please correct … synthetic
Line 95: please correct … cell wall
Line 125: please correct … purchased
Line 127: please correct … 100 mbar
Line 153: …were ensured.
Fig 2: please correct… reprezentation…rezidual….
Lines 179, 230: Saccharomyces pastorianus should be written in italic
Line 191 ff: …For experimental data modelling, three of the most known biosorption equilibrium
models: Freundlich (F), Langmuir (L) and Dubinin – Radushkevich (D-R), shortly synthesized (what does it mean in the given context?) below, were applied [12].
Line 270: …peak?..
Fig. 6, and line 329 ff: … composite material based on residual Saccharomyces pastorianus residual biomass encapsulated…
Line 316: Gill‘s classification instead of Giles
Fig 7: graph b is wrong. Please check it. It seems to be graph f as double.
Tab 1: values for d and f for T1 (5 °C) are missing.
Ref. [18] and [19] are misssing in the text.
The term Penicilium is orthographically not correct, but used in [20]
The following paragraph should be carefully revised and translated into correct sentences:
Thus, using Penicilium sp immobilized in 2% sodi-387 um alginate was removal C. I. Reactive Red dye (q= 120.48 mg/g) [20]; Lentinus concinnus 388 biomass immobilized to carboxymethyl cellulose (CMC), in the presence of FeCl3 was 389 removal Disperse Red 60 dye ( q = 92.6 mg/g) [21]; Lentinus concinnus biomass immobi-390 lized in polyvinyl alcohol /polyethyleneoxide hydrogels was removal Reactive Yellow 86 391 dye (q=87.6 mg/g) [22]; Bacillus subtillis sp. immobilized in sodium alginate was removal 392 Brilliant Red HE-3B reactive dye (q = 588.235) [23]; Lactobacillus sp. immobilized in so-393 dium alginate was remova; Orange 16 (q = 123. 459 mg/g)
Line 400: please correct … achieved.
Line 405: Table 3 is missing. Do the authors mean Table 2???
Discussion of the value for ΔS is missing.
Line 421: bioamendments, please correct.
Author Response
Dear Reviewer #1,
We thank you for your time spent for reviewing our manuscript and for your comments and suggestions that have been helpful to improve its quality.
We are happy that you appreciated our manuscript and considered that it has ‘an interesting subject which is well covered by the authors’.
We revised our manuscript and considered all your recommendations. All changes made in the manuscript were highlighted in red.
Reviewer Suggestions/Recommendations (R1C)
Authors answer (A1A)
R1C 1 How, for example, did the authors verify the complete inactivation of the yeasts by drying?
AA 1. The residual Saccharomyces pastorianus yeast was inactivated by drying for 4 hours at 80°C (row 146). The drying process at high temperature inactivates the yeast and improves the biosorption capacity by protein denaturation. Temperature above 60°C results in complete loss of cells viability [Valentine, G.D.S.; Walker, M.E.; Gardner, J.M.; Schmid, F.; Jiranek, V. Brief temperature extremes during wine fermentation: Effect on yeast viability and fermentation progress. Aust. J. Grape Wine Res. 2018.https://doi.org/10.1111/ajgw.12365].
R1C 2. Was the biomass used characterised in more detail, for example the tendency to form spores?
AA 2. The strains Saccharomyces pastorianus does not have the ability to form spores [Table 2 from .A. Harrison, Beer/Brewing,Ed. Moselio Schaechter, Encyclopedia of Microbiology (Third Edition), Academic Press,2009, Pages 23-33, ISBN 9780123739445, https://doi.org/10.1016/B978-012373944-5.00117-6 ]
R1C 3. Furthermore, several orthographic and grammatical errors or incorrectnesses are conspicuous in the text, even for the linguistically unskilled reader. The presentation of results needs to be revised, as does the list of literature references. The following comments are intended to help with this:
Line 33: please correct … microcapsules of about 1500 µm
Modification was made in the text
Line 75: please correct … synthetic
Modification was made in the text
Line 95: please correct … cell wall
Modification was made in the text
Line 125: please correct … purchased
Modification was made in the text
Line 127: please correct … 100 mbar
Modification was made in the text
Line 153: … were ensured.
Modification was made in the text
Fig 2: please correct… reprezentation…rezidual….
Modification was made in the text
Lines 179, 230: Saccharomyces pastorianus should be written in italic
Modification was made in the text
Line 191 ff: …For experimental data modelling, three of the most known biosorption equilibrium
models: Freundlich (F), Langmuir (L) and Dubinin – Radushkevich (D-R), shortly synthesized (what does it mean in the given context?) below, were applied [12].
In the scientific literature there is a wide variety of equilibrium isotherm models such as Langmuir, Freundlich, Brunauer–Emmett–Teller, Redlich–Peterson, Dubinin–Radushkevich, Temkin, Toth, Koble–Corrigan, Sips, Hill. We chose only three, which we considered more relevant for the subject, selected according to the experimental data, the biosorption system and the experience of the team in modeling the adsorption equilibria in the case of different systems.
Line 270: …peak?..
Modification was made in the text
Fig. 6, and line 329 ff: … composite material based on residual Saccharomyces pastorianus residual biomass encapsulated…
Modification was made in the text
Line 316: Gill‘s classification instead of Giles
Modification was made in the text
Fig 7: graph b is wrong. Please check it. It seems to be graph f as double.
Sorry for mistake. Modification was made in the Figure 7a
Tab 1: values for d and f for T1 (5 °C) are missing.
The studied system has an anti-Langmuir behaviour at a temperature of 278 K, reason for which the characteristic parameters in this case could not be calculated. Their values would have given a negative result, as can be seen from the position of the line characteristic of this temperature in Figs. 7 (c, d,e,f).
Ref. [18] and [19] are misssing in the text.
Ref were added in the text:
For the removal of Reactive Red 198 and Reactive Yellow 2 dyes a microbial cell-immobilized Platanus orientalis leaf tissue was used and biosorption capacities of 51.12 and 37.59 mg/g were recorded [26]. Immobilized Mucor plumbeus on sepiolite support was used for Methyl violet biosorption with 187.76 mg/g biosorption capacity [27].
The term Penicilium is orthographically not correct, but used in [20]
Modification was made in the text
The following paragraph should be carefully revised and translated into correct sentences:
Thus, using Penicilium sp immobilized in 2% sodi-387 um alginate was removal C. I. Reactive Red dye (q= 120.48 mg/g) [20]; Lentinus concinnus 388 biomass immobilized to carboxymethyl cellulose (CMC), in the presence of FeCl3 was 389 removal Disperse Red 60 dye ( q = 92.6 mg/g) [21]; Lentinus concinnus biomass immobi-390 lized in polyvinyl alcohol /polyethyleneoxide hydrogels was removal Reactive Yellow 86 391 dye (q=87.6 mg/g) [22]; Bacillus subtillis sp. immobilized in sodium alginate was removal 392 Brilliant Red HE-3B reactive dye (q = 588.235) [23]; Lactobacillus sp. immobilized in so-393 dium alginate was remova; Orange 16 (q = 123. 459 mg/g)
The paragraph has been replaced with:
Penicillium sp. immobilized in 2% sodium alginate was used for the removal of C. I. Reactive Red dye obtaining a maximum sorption capacity of 120.48 mg/g [28]. Lentinus concinnus biomass immobilized to carboxymethyl cellulose (CMC), in the presence of FeCl3 achieved the removal Disperse Red 60 dye with a maximum q value of 92.6 mg/g [29]. The same strain (Lentinus concinnus) immobilized in polyvinyl alcohol /polyethylene oxide hydrogels was employed for the retention of Reactive Yellow 86 dye with a biosorption capacity of 87.6 mg/g [30]. Bacillus subtillis sp. immobilized in sodium alginate was used for Brilliant Red HE-3B reactive dye biosorption obtaining 588.235 mg/g maximum sorption capacity [31]. Lactobacillus sp. immobilized in sodium alginate successfully removed Orange 16 dye from aquous solutions with a q = 123. 459 mg/g value [32].
Line 400: please correct … achieved.
Modification was made in the text
Line 405: Table 3 is missing. Do the authors mean Table 2???
Yes. Modification was made in the text
Discussion of the value for ΔS is missing.
It was added in the text.
Line 421: bioamendments, please correct.
Modification was made in the text
Sincerely yours,
Reviewer 2 Report
A new proposed polymeric composite based on residual microbial biomass encapsulated in sodium alginate. The polymer was fully characterized, and the batch experiment in the study of the biosorption process, such as the dose of biosorrbent, the size of the composites granules, the pH of the solution and the initial concentration etc. were investigated. The adsorption equilibrium isotherm was also studied to estimate the characteristic parameters for the application in dye-containing watery effluents. The research is interesting and meanningful. However, some explainations are not very clear. I suggest major revise of the manuscript.
1. The introduction should be rewrite, because the author do not give a fully introduction on the use of microorganisms in environmental pollutant removal, especial for the microorganisms immobilization on different polymers. How about the removal efficiency and what's the bottleneck for the encapsulate of microorganisms techniques? A comparative analysis of the literatures must be available.
2. How to test the activity of the Saccharomyces pastorianus ? How about the water solubility of the biosorbent? Does this affect the adsorption results?
3. Will the biosorbent break or mass loss during use? How to recycle the biosorbent and what's the re-use efficiency? The reuse experiment should be added.
4. For the physical-chemical characterization of the prepared polymeric composite, more analysis should be provided, for example the BET, XPS, etc. Because the author claimed that the biosorption process based on a physical adsorption mechanism. Therefore, the specific surface area of the biosorbent should be provided. More details can be found in the Ref. Sun et al., Biores. Technol. 344, 126186.
Author Response
Dear Reviewer #2,
We thank you for your time spent for reviewing our manuscript. Sorry that you have found our manuscript without sufficient data to support title or without a detailed and critical review provided.
We revised our manuscript and considered your specific comments. All changes in manuscript text are marked with track changes.
Reviewer specific comments (R2C)
Authors answer (AA)
R2C 1. The introduction should be rewrite, because the author do not give a fully introduction on the use of microorganisms in environmental pollutant removal, especial for the microorganisms immobilization on different polymers. How about the removal efficiency and what's the bottleneck for the encapsulate of microorganisms techniques? A comparative analysis of the literatures must be available
AA 1 The folowing paragraphs were added:
Different immobilisation techniques can be applied to microbial biomass: entrapment, cross-linking, covalent bonding, adsorption and encapsulation. The main problem related to the use of microencapsulation for biosorbents production is the limitations imposed by the diffusion step in the pollutants retention [4].
Different microorganisms have been used successfully in environmental pollutant removal:
- bacteria: Pseudomonas aeruginosa immobilised in sodium alginate was used for the retention of Reactive green 6 from wastewaters with a maximum adsorption capacity of 21.2 mg/g; Bacillus, immobilized in 1% sodium alginate allowed obtaining a maximum adsorption capacity of 588.235 mg/g for Briliant Red-HE-3B; Bacillus cereus immobilised in 3% sodium alginate yielded a maximum retention of 83% for Malachite green [10-12].
- fungi: Penicilium immobilised in 2% sodium alginate was used for the removal of C.I.Reactive Red with a maximum adsorption capacity of 120.48 mg/g; Penicilium crustosum immobilised in 2% agar retained Congo red with an efficiency of 81.86%; Rhizophus orizae immobilised in carboxymethyl cellulose was used for the biosorption of Reactive Blue with an efficiency of 97.44%. Saccharomyces cerevisiae immobilized in sodium alginate and present in the form of gel beads was also used for the retention of Brilliant Red HE-3B dye and led to a sorption capacity of 104.67 mg/g [13-16].
Khashei et al investigated the biosorption ability for heavy metals of immobilized Pseudomonas putida cells in various matrices (alginate–PVA–CaCO3 and carboxymethyl cellulose). An increase of metal removal efficiency in all matrices after bacterial immobilization was observed: 75.5% Pb(II) compared to 60% and 75% Cd(II) to 20% without cells, for alginate–PVA–CaCO3, and 32% Pb(II) retention to 3% and 15% Cd(II) to 5%, for cellulose support [18]
R2C 2. How to test the activity of the Saccharomyces pastorianus ? How about the water solubility of the biosorbent? Does this affect the adsorption results?
AA 2. The biosorption implies an inactive biosorbent, thus the residual yeast used in experiments was inactivated by drying for 4 hours at 80°C (row 146). The drying process at high temperature inactivates the yeast and improves the biosorption capacity by protein denaturation. The obtained biosorbents are water insoluble – the microencapsulation implies the confinement of microbial biomass within a semipermeable polymeric matrix that enables the cells physical isolation from the external environment while maintaining their biosorptive abilities. Temperature above 60°C results in complete loss of cells viability [Valentine, G.D.S.; Walker, M.E.; Gardner, J.M.; Schmid, F.; Jiranek, V. Brief temperature extremes during wine fermentation: Effect on yeast viability and fermentation progress. Aust. J. Grape Wine Res. 2018.https://doi.org/10.1111/ajgw.12365].
R2C 3. Will the biosorbent break or mass loss during use? How to recycle the biosorbent and what's the re-use efficiency? The reuse experiment should be added.
AA 3. No modifications in biosorbent structure were noticed during the experiments. This type of biosorbent is not reused. Due to the gelation structure, the biosorbent cannot be reused and can be used by other known methods (reference 37): composting, anaerobic digestion or soil amendment. In subchapter 3.5 of this manuscript a brief indication is made on this subject. We have applied this last method, and the results are captured in a scientific paper that is being reviewed (reference 36). They were also presented at the international conference: Life Science Today for Tomorrow, International Congress, 21-22 October 2021, Iasi, Romania (Tataru-Farmus R.E., Zaharia C., Suteu D., Blaga A.C., Characteristics and wheat grains development trends)
R2C 4. For the physical-chemical characterization of the prepared polymeric composite, more analysis should be provided, for example the BET, XPS, etc. Because the author claimed that the biosorption process based on a physical adsorption mechanism. Therefore, the specific surface area of the biosorbent should be provided. More details can be found in the Ref. Sun et al., Biores. Technol. 344, 126186.
AA 4. Thank you very much for the suggested bibliographic indication. It is very valuable for our studies on lignocellulosic materials. In this case, however, it is a biosorbent with a gelatinous structure, which is stored until use in calcium chloride. In order to be analyzed by physico-chemical methods (SEM, FTIR) these materials were initially dried by lyophilization (which we also specified in the manuscript, pg.6, lines 198 - 202). Determining the specific surface area by the BET method would not be relevant because the method will be applied to a different material in terms of physical properties (moisture, porosity) than that used in biosorption experiments. These polymeric composites are different from lignocellulosic biomass (solid material, with controlled stability and humidity). We resorted to those methods that could provide us with valuable information for this type of biosorbent.
Sincerely yours,
Reviewer 3 Report
The work presents a good scientific standard.
My comments are as follows:
- Page 1, lines 41-42; page 14 line 377; page 15, line 409; 2; conclusions; etc. - unit kJ/mol
- The introduction should be completed. Poor literature review. Complete information on Brilliant Red HE-3B. Why was this dye chosen in the biosorption experiments?
- Page 8 , lines 240 -242 I quote: ,,Porosity and large surface area are very important elements in the case of materials used as adsorbent for chemical pollutants”. Structural parameters should be determined and the results obtained in the study should be completed in this work (nitrogen adsorption/ desorption isotherms).
- Page 9 ,lines 276 , 279 etc. , In the symbol of point of zero charge, I suggest PZC in subscript, this applies to the markings throughout the work. This method is not mentioned in the abstract, it should be supplemented.
- Page 15 , line 405 , the data of free energy Gibbs, ΔG0 are presented in table 2, not table 3
- Page 15 , line 432 underlined Radushkevich - mistake
Author Response
Dear Reviewer #3,
We thank you for your time spent for reviewing our manuscript and for your comments and suggestions that have been helpful to improve its quality.
We revised our manuscript and considered all your recommendations. All changes made in the manuscript were highlighted in red.
Reviewer Suggestions/Recommendations (R3C)
Authors answer (AA)
R3C 1. Page 1, lines 41-42; page 14 line 377; page 15, line 409; 2; conclusions; etc. - unit kJ/mol
AA 1. Modification was made in the text
R3C 2. The introduction should be completed. Poor literature review. Complete information on Brilliant Red HE-3B. Why was this dye chosen in the biosorption experiments?
AA 2. A number of changes have been made to the text of the input displayed on the text with track changes
- Different immobilisation techniques can be applied to microbial biomass: entrapment, cross-linking, covalent bonding, adsorption and encapsulation. The main problem related to the use of microencapsulation for biosorbents production is the limitations imposed by the diffusion step in the pollutants retention [4].
Different microorganisms have been used successfully in environmental pollutant removal:
- bacteria: Pseudomonas aeruginosa immobilised in sodium alginate was used for the retention of Reactive green 6 from wastewaters with a maximum adsorption capacity of 21.2 mg/g; Bacillus, immobilized in 1% sodium alginate allowed obtaining a maximum adsorption capacity of 588.235 mg/g for Briliant Red-HE-3B; Bacillus cereus immobilised in 3% sodium alginate yielded a maximum retention of 83% for Malachite green [10-12].
- fungi: Penicilium immobilised in 2% sodium alginate was used for the removal of C.I.Reactive Red with a maximum adsorption capacity of 120.48 mg/g; Penicilium crustosum immobilised in 2% agar retained Congo red with an efficiency of 81.86%; Rhizophus orizae immobilised in carboxymethyl cellulose was used for the biosorption of Reactive Blue with an efficiency of 97.44%. Saccharomyces cerevisiae immobilized in sodium alginate and present in the form of gel beads was also used for the retention of Brilliant Red HE-3B dye and led to a sorption capacity of 104.67 mg/g [13-16].
Khashei et al investigated the biosorption ability for heavy metals of immobilized Pseudomonas putida cells in various matrices (alginate–PVA–CaCO3 and carboxymethyl cellulose). An increase of metal removal efficiency in all matrices after bacterial immobilization was observed: 75.5% Pb(II) compared to 60% and 75% Cd(II) to 20% without cells, for alginate–PVA–CaCO3, and 32% Pb(II) retention to 3% and 15% Cd(II) to 5%, for cellulose support [18]
- Brilliant Red HE-3B is a reactive dye specific to dyeing cellulose fibers. It is a dye used by us as a model in biosorption or sorption studies on various adsorbent materials, conventional or unconventional.
R3C 2. Page 8 , lines 240 -242 I quote: ,,Porosity and large surface area are very important elements in the case of materials used as adsorbent for chemical pollutants”. Structural parameters should be determined and the results obtained in the study should be completed in this work (nitrogen adsorption/ desorption isotherms).
AA 2. It is a biosorbent with a gelatinous structure, which is stored until use in calcium chloride. In order to be analyzed by physico-chemical methods (SEM, FTIR) these materials were initially dried by lyophilization (which we also specified in the manuscript, pg.6, lines 198 - 202). Determining the specific surface area by the BET method would not be relevant because the method will be applied to a different material in terms of physical properties (moisture, porosity) than that used in biosorption experiments. These polymeric composites are different from other solid biomass (i.e. lignocellulosic - solid material, with controlled stability and humidity). We resorted to those methods that could provide us with valuable information for this type of biosorbent.
R3C 3. Page 9 ,lines 276 , 279 etc. , In the symbol of point of zero charge, I suggest PZC in subscript, this applies to the markings throughout the work. This method is not mentioned in the abstract, it should be supplemented.
AA 3. Changes have been made to the text
R3C 4. Page 15 , line 405 , the data of free energy Gibbs, ΔG0 are presented in table 2, not table 3
AA 4. Modification was made in the text
R3C5. Page 15 , line 432 underlined Radushkevich – mistake
AA 5. Modification was made in the text
Sincerely yours
Round 2
Reviewer 1 Report
The manuscript can be accepted in the present form.
Reviewer 2 Report
The comments have been answered, and the paper can be accept in the present form.